# Probing Spatiotemporal Effects of Intertrack Recombination with a New Implementation of Simultaneous Multiple Tracks in TRAX-CHEM

**DOI:** 10.3390/ijms26020571

**Published:** 2025-01-10

**Authors:** Lorenzo Castelli, Gianmarco Camazzola, Martina C. Fuss, Daria Boscolo, Michael Krämer, Valentina Tozzini, Marco Durante, Emanuele Scifoni

**Affiliations:** 1Department of Physics, University of Trento, 38121 Trento, Italy; lorenzo.castelli-1@unitn.it; 2Trento Institute for Fundamental Physics and Application, TIFPA, 38123 Povo, Italy; 3Istituto Nanoscienze-CNR, NEST-SNS, 56127 Pisa, Italy; 4Biophysics Division, GSI Helmholtzzentrum für Schwerionenforschung GmbH, 64291 Darmstadt, Germany; 5Department of Medical Physics, MedAustron, 2700 Wiener Neustadt, Austria; 6INFN, 56127 Pisa, Italy

**Keywords:** radiation track structure simulation, FLASH radiotherapy, ultra-high dose rate response, intertrack recombination

## Abstract

Among the most investigated hypotheses for a radiobiological explanation of the mechanism behind the FLASH effect in ultra-high dose rate radiotherapy, intertrack recombination between particle tracks arriving at a close spatiotemporal distance has been suggested. In the present work, we examine these conditions for different beam qualities and energies, defining the limits of both space and time where a non-negligible chemical effect is expected. To this purpose the TRAX-CHEM chemical track structure Monte Carlo code has been extended to handle several particle tracks at the same time, separated by pre-defined spatial and temporal distances. We analyzed the yields of different radicals as compared to the non-interacting track conditions and we evaluated the difference. We find a negligible role of intertrack for spatial distances larger than 1 μm, while for temporal distances up to μs, a non-negligible interaction is observed especially at higher LET. In addition, we emphasize the non-monotonic behavior of some relative yield as a function of the time separation, in particular of $H_{2}O_{2}$, due to the onset of a different reaction involving solvated electrons besides well-known $OH^{\cdot }$ recombination.

## 1. Introduction

The cascade of physical, chemical, and biological events caused by irradiating living matter with ionizing particles is the subject of intense investigation [1], both for the aim of using them as cancer treatment, that is radiotherapy, and to prevent the pathological side effects and long-term consequences induced by undesired radiation exposure. The former relies on exploiting the intrinsic physical interaction features of different radiation types, mainly photons, electrons, protons, carbon, and helium ions, to induce tumor cell killing. However, the radiotherapy application itself is constrained by the possible damage induction to healthy tissues surrounding the (cancerous) target [2,3]. To mitigate these drawbacks, conventional radiotherapy is usually delivered in multiple treatment sessions spread over several weeks which however, does not completely prevent the risk of damage to healthy tissues manifesting as late-onset side effects like fibrosis, impaired organ function, necrosis, and secondary malignancies. An emerging treatment approach called FLASH radiotherapy (FLASH-RT [4,5]) offers a potential breakthrough in mitigating these challenges. FLASH-RT stands out for its ultrafast radiation delivery, with average dose rates surpassing those of conventional radiotherapy by several orders of magnitude (typically >40 Gy/s compared to 0.5–5 Gy/min, respectively) [6]. There is increasing evidence [7,8,9,10,11] that this new dose delivery modality can spare significantly and selectively healthy tissues at the same therapeutic effects on the tumor, a feature known as the “FLASH effect” [12,13].

Despite the potential of FLASH-RT, its clinical translation is still in its early phases. The effects of FLASH-RT are currently already being examined through initial clinical trials ([14,15]), while preclinical experimental studies both in vitro and in vivo (see e.g., [16,17,18,19]), complemented by extensive modeling and simulation studies (see e.g., [20,21,22]) are continuously attempting to provide insights into their understanding. Still, indeed, the underlying cellular mechanisms driving the differential impact on cancerous vs normal tissues, and why these effects are not observed in conventional radiotherapy, remain largely unknown. Several aspects have been proposed to be at the basis of the FLASH effect, including variations in tumor tissue organization and vascularization [23], disparities in cellular and sub-cellular structure and biochemistry [24], even extending to fundamental molecular-level discrepancies in post-irradiation chemistry [21,25]. At this level, various hypotheses, neither confirmed nor completely refuted, speculate, e.g., about the role of differing dissolved oxygen concentrations in healthy versus cancer cells [20,21,26].

One hypothesis suggests that inter-track interactions, involving reactions between chemicals produced from different primary particle tracks, may play a key role, already in the heterogeneous early radiolysis phase. Previous Monte Carlo track structure (TS) simulation studies using protons, carbon, and helium ions with PARTRAC [27,28], and using protons with TOPAS-nBio [29,30], have explored radical-radical interactions and the yields of chemical radicals induced by multiple projectiles of the same radiation quality on the same target, computed for varying linear energy transfer (LET). Different and sometimes diverging results were observed for the inter-track effect at high dose rates [22,30,31,32,33], indicating the need for further examination.

The present research also extends to scenarios where the FLASH effect is not observed but where similar spatiotemporal dynamics are at play. For instance, ultra-bunched irradiations generated by laser-driven particle beams present elevated intrapulse dose rates due to the close spatiotemporal proximity of particles [34,35,36]. However, these setups’ dose per pulse and repetition frequency typically do not achieve the macroscopic average dose rates required to induce a FLASH effect [37]. By addressing these irradiation conditions, our study underscores its broader relevance, offering insights into radiation effects that are not limited to FLASH radiotherapy but extend to novel irradiation modalities.

In this context, in previous works, we enhanced and developed the chemical extension of TRAX [38], called TRAXCHEM [39], and dedicated further extensions [40] to study several features connected to the FLASH mechanism. By explicitly modeling the interaction between radiation tracks and target oxygenation [41,42], we emphasized for the first time the inconsistency of the oxygen depletion hypothesis, which was later confirmed by extensive experimental data [18,43]. Recently we also extended the temporal evolution of the tracks up to ms timescales, through the analytical dedicated extension TRAXCHEMxt [40], which uses the distribution of chemical species from TRAXCHEM at a transition time of approximately 500 ns as input.

In the present work, we further extend our code to allow the evolution and interactions of simultaneous multiple tracks, separated by different time and space values. The main goal of this study is to scrutinize intertrack conditions for different particle types and energy, by defining the limits of spatiotemporal proximity where non-negligible effects emerge. The implementation of inter-track interactions in the simulations reveals the ranges of conditions in which different types of reactions between different tracks may have a non-negligible impact on modifying the overall yields of damaging chemical species, thus contributing to a dose rate-dependent protective effect. In the materials and methods section, we provide the technical details of the code along with a general overview of the irradiation conditions and geometry. In the results and discussion sections, we analyze the impact of intertrack interactions in comparison with previous studies and examine the limitations of the current assessment of this effect.

## 2. Results

### 2.1. Spatiotemporal Separation Between Individual Tracks: p, He, C

G-values were computed for pairs of proton, helium, and carbon ion tracks with various spatial and temporal separations and compared to those without intertrack effects (NI), where the two tracks are processed entirely independently of each other (i.e., Δx →∞). See Section 4, Materials and Methods, for G-values definition and simulation details.

Time-dependent G-values of $OH^{\cdot }$ and $H_{2}O_{2}$ produced by carbon ions are depicted in Figure 1. The first column shows the curves for fixed spatial distance, while the second column shows the behavior for a fixed time delay between the particles. G-values for an ion-track pair with a time delay Δt in between show a sudden change when the second ion track enters the volume of interest. At the arrival time of the second track, the G-value can change in two different ways depending on the observed chemical species. If we are studying primary radiolytic products, i.e., species generated through direct interaction of radiation with water such as $OH^{\cdot }$, $e_{aq}^{-}$, and $H_{3}O^{+}$, then the curve has a peak (first row). Conversely, if the focus is on species mainly generated by further reactions (secondary radiolytic products) like $OH^{-}$, $H_{2}O_{2}$, $H_{2}$, and $H^{\cdot }$, then the curve has a drop (second row). The value of the peak/drop for a chosen radical at a fixed Δt is the same for all combinations of Δx because the G-value at the peak/drop corresponds to the mean between the G-value at the ps, i.e., at the start of the chemical stage, right after the prechemical end, and the G-value at Δt (1)$$\mathrm{G-value}\left(\Delta t\right)=\frac{N_{1}\left(\Delta t\right)+N_{2}\left(\mathrm{ps}\right)}{E_{1}+E_{2}}=\frac{N(\Delta t)+N(\mathrm{ps})}{2E}≷\mathrm{G-value}\left(\mathrm{ps}\right).$$In Equation (Equation 1), $N_{i}$ and $E_{i}$ represent respectively the number of radicals generated from, and the energy deposited by, the first and second track. The second equation holds as the values of *N* and *E* at the arrival of the second track can be considered as mean across independent NI simulations since the intertrack effect is not yet active: the first track evolved independently of the second up to Δt, while the latter physical and prechemical stages are by definition independent of the first track. The choice between ≷ depends on the species under analysis. Primary products display a peak (Figure 1, $OH^{\cdot }$) lower than the value at 1 ps due to their reduced number at Δt resulting from intra-track interactions. For secondary species, a drop is observed since no molecule is generated upon the arrival of the second track. Unlike the previous situation, the drop value is greater than that at 1 ps, and it increases with time delay Δt, following the behavior of the G-value as illustrated in Figure 1 by $H_{2}O_{2}$ evolution.

Following the peak/drop, two distinct behaviors emerge. Primary species experience depletion, with the G-value consistently lower than that of NI, indicating increased consumption. In contrast, the secondary species’ G-value is mostly enhanced by intertrack effects, signifying higher production (with some exceptions discussed in the next section). The relationship between the intertrack effect and the spatial and temporal distance between tracks is evident from Figure 1. When tracks arrive with the same time delay Δt but different spatial distances, the effect becomes more pronounced as the distance between tracks decreases. Reducing the distance results in a larger deviation from the NI G-value, irrespective of the type of radicals studied, while enlarging it makes the G-value converge to NI. Similarly, for different time delays, the impact varies depending on the arrival time of the second particle; a shorter delay leads to a greater impact. In general, for small Δt and Δx, the G-value curve follows a behavior expected for identical ion types with rising LET.

### 2.2. Intertrack Effects with Different LETs

To assess the significance of the impact of spatial and temporal overlap at different LETs, the calculated G-values were examined after 1 μs from the arrival of the second particle as a function of Δt and Δx for each radiation quality and chemical species. Each bin in the two-dimensional histograms (Figure 2) represents the normalized difference between the G-value obtained under a specific irradiation condition and the G-value at time 1μs +Δt in the NI scenario (The value at 1μs+Δt for NI is: ${\mathrm{G-value}}_{\mathrm{NI}}=\left({\mathrm{G-value}}_{\mathrm{NI}}\left(1\;\mu s+\Delta t\right)+{\mathrm{G-value}}_{\mathrm{NI}}\left(1\;\mu s\right)\right)/2$ )(2)$$\Delta {\mathrm{G-value}}_{\mu s}=\frac{\mathrm{G-value}\left(1\;\mu s+\Delta t\right)-{\mathrm{G-value}}_{\mathrm{NI}}}{{\mathrm{G-value}}_{\mathrm{NI}}}\cdot$$ This approach allows us to observe quantitatively the effect of the intertrack overlap across different combinations of Δx and Δt for each species. In Figure 2, the color of each bin indicates the difference in G-value: red represents a positive difference, blue indicates a negative difference, and shades toward white signify values closer to the NI scenario.

For protons, shown in the first column of Figure 2, the effect on molecules directly generated by radiation interactions, such as $OH^{\cdot }$, $H_{3}O^{+}$, and $e_{aq}^{-}$, consistently varies with spatial and temporal distance (see Appendix A for $H_{3}O^{+}$). Notably, the disparity between NI and intertrack scenarios is most pronounced for closely spaced and timed projectiles. Radicals are depleted at a higher rate compared to the NI, with values at 1 μs consistently lower for each space-time combination. For all three chemical species, the deviation from the NI scenario tends to decrease with increasing spatial and temporal distance. A significant correlation can be observed between the depletion effect and spatial distance, where the effect is notably reduced beyond Δx =10 nm. Similarly, as temporal distance increases, the depletion effect gradually weakens but remains noticeable even at Δt =1μs. At this point, for each Δx, the effect is moderate, and for Δx $=10^{3}$ nm, it becomes comparable to the NI scenario (under the T-Student test). These observed effects can be compared with results for $OH^{\cdot }$ in PARTRAC [28]. While the overall behavior aligns with previous observations, a significant difference is noted at Δx =1000 nm, where Kreipl suggested, regardless of Δt, no difference between intertrack and NI scenarios, whereas here a non-negligible effect is observed (for more info see Appendix A). This difference is likely due to variations in the cross-sections used in the physical stage in TRAX and PARTRAC, as well in the prechemical branching ratios (see [39] for differences between TS codes on this level), resulting in a more laterally spread track for the former Monte Carlo code.

A similar but opposite effect is observed for chemical species produced by subsequent reactions, such as $H_{2}O_{2}$ and $H_{2}$. These species are produced to a greater extent compared to the NI case due to the fast and numerous reactions of primary (radiolytic) species. As for directly produced radicals, the effects are more pronounced for projectiles at close proximity in time or space. So far the effect of multiple tracks resembles the result of increasing the LET for radiation. Here, the increase in projectiles leads to the generation of a greater quantity of chemical species that interact with each other leading to depletion or enhancement similar to what a greater deposition of energy would do.

In contrast, $H^{\cdot }$ exhibits different behavior, characterized by lower deviations and the appearance of two distinct regions for delays smaller or larger than Δt=10 ns. If the second track arrives before 10 ns, the intertrack effect enhances radical production, likely due to the mutual consumption of $e_{aq}^{-}$ and $H_{3}O^{+}$, which are the primary species responsible for producing $H^{\cdot }$ in the presence of water: (3)$$H_{3}O^{+}+e^{-}\to H^{\cdot }+H_{2}O.$$ This enhancement contradicts what one might expect from a simple LET increase and is likely due to the relatively low density of $H^{\cdot }$ radicals, whose reaction probability with other molecules is lower than that for the process in Equation (Equation 3). As the temporal distance increases, this enhancement becomes less significant because within the ns $e_{aq}^{-}$ and $H_{3}O^{+}$ are nearly halved reducing their ability to react efficiently. Consequently, the environment generated by the arrival of the first track is no longer able to provide a density high enough to produce $H^{\cdot }$, on the contrary, the second track finds an environment such that their consumption is favored (Equation (Equation 4)).(4)$$\begin{array}{c} \;\;\;e^{-}+H^{\cdot }\to H_{2}+OH^{-}\\ OH^{\cdot }+H^{\cdot }\to H_{2}O\\ \;\;\;H^{\cdot }+H^{\cdot }\to H_{2}. \end{array}$$

By varying LET and radiation quality, some differences between the histograms arise, in Figure 2 each column corresponds to a different radiation quality and each row to a specific chemical species. Overall, the histograms of $OH^{\cdot }$, $H_{3}O^{+}$, and $e_{aq}^{-}$, closely resemble those for protons: they exhibit a depletion that intensifies as the spatial and temporal proximity between projectiles increases. However, the extent of depletion differs; notably, with increasing LET, the depletion for helium and carbon is nearly double that observed with protons. On the other hand, the behavior at Δx =1000 nm is different. Here, the depletion of primary products quickly approaches zero (near NI) at high Δt, as the tracks formed at higher LET are denser, implying a more challenging interaction at greater distances and a higher number of reactants formed at close spatial distance between tracks.

Similarly, a comparable behavior is observed for $H_{2}$, which is enhanced by the intertrack effect following similar patterns observed for protons. However, at Δx =1000 nm, it fast approaches the NI scenario as Δt increases.

Conversely, $H^{\cdot }$ radicals show signs of depletion with increasing LET, as expected. The exception occurs at Δx=1000 nm, where a slight enhancement is observed, likely due to an unexpected production reaction that becomes significant at low Δt. A possible explanation for this enhancement is that, at longer distances (>500 nm), the chemical environment is predominantly composed of $e_{aq}^{-}$ and $H_{3}O^{+}$, which can react according to Equation (Equation 3) in order to produce $H^{\cdot }$.

Lastly, $H_{2}O_{2}$ presents an interesting behavior. Similar to $H^{\cdot }$ in proton tracks, $H_{2}O_{2}$ exhibits a threshold at $\Delta t=10^{5}$ ps, where it transitions from enhancement to depletion due to its reaction with $e_{aq}^{-}$: (5)$$H_{2}O_{2}+e^{-}\to OH^{-}+OH^{\cdot }.$$ This depletion effect is observed at different degrees both in carbon and helium simulations, and it’s attributed to the environment generated by the first track. In this scenario, $H_{2}O_{2}$ becomes distributed within a region where the second track generates reactive species, such as $e^{-}$, which efficiently consume them.

In summary, the differences observed between the normalized yields of protons and carbon ions are also evident for protons vs. helium, with the discrepancies increasing as a consequence of the latter’s higher LET.

## 3. Discussion

We conducted a quantitative analysis of the variation in G-values for several species in close spatiotemporal proximity across different particle tracks and LET. Our findings show that most interactions become negligible at the boundaries of the configuration space (Δt =1
μs and Δx =1
μm). As observed by Kreipl [28], there is a region where a substantial (>10%) change in G-values is observed, corresponding to low Δt and Δx between tracks. This effect aligns with the track structure shown in Figure 3, where most chemical species are located within 100 nm of the track center up to 1 μs.

At Δx =1
μm, we observed an unexpected behavior compared to the previous studies [28,30,33]. For protons, all chemical species exhibit a small but statistically significant intertrack effect, which only approaches the NI scenario at Δt =1
μs. The only exception is $H^{\cdot }$ which remains very low across all Δx and Δt combinations. Helium and carbon ions show a similar pattern, but their effects are significant only for Δt <1 ns due to the rapid consumption of radicals during chemical evolution. These results can be explained by the broader track structure of TRAX-CHEM compared to other codes. As shown in Figure 4, the number of species outside 500 nm is still sufficient to influence the chemical evolution of the track. Additional tests demonstrated that, regardless of the primary particle or Δt, tracks at Δx >1.5
μm do not exhibit intertrack interactions.

Although this observed effect resembles LET enhancement, as seen in the other works [28,30,33], our simulations reveal that this is not consistent across all chemical species analyzed. Notable exceptions include $H^{\cdot }$ at low LET (protons) and $H_{2}O_{2}$ at high LET (carbon and helium). This divergence can be attributed to changes in the chemical environment encountered by the second track. Figure 3 illustrates how the proportion of chemical species shifts over time, particularly for helium and carbon, where the rapid depletion of primary radiolytic products leads to the formation of secondary products. As a result, the second track develops in a chemically distinct environment, which alters the intertrack effect—specifically, the consumption of first track $H_{2}O_{2}$ by the solvated electrons from the second track.

Depletion of $H_{2}O_{2}$ at high Δt was also observed for 40 MeV/u carbon ions (see Appendix A), which have a LET close to 30 keV/μm (similar to helium ions), further confirming that this trend can be triggered by an increase in the projectile’s LET.

At high Δt, the effects are modest due to the reduced number of available chemical species (Figure 3), but they remain significant. These qualitative and quantitative differences in the intertrack effect highlight the challenges in modeling this phenomenon using the assumption of geometric overlap between simultaneous tracks, as such an approach neglects the density and chemical composition changes within the track.

In the present analysis, we did not account for the role of oxygen and scavengers, even if both contribute to the chemical evolution of the track and so could potentially alter the intertrack effect. Preliminary estimates performed with the feature already available in TRAXCHEM [41] lead to a minor effect due to oxygen only, for a moderate concentration resembling physioxic levels (7% pO_2_), Detailed analysis of oxygen will be considered in future investigations, as well as scavengers as it is necessary to fully investigate the role of intertrack interactions.

A significant limitation in ultra-high dose rate radiation chemistry is the scarcity of experimental data at the sub-microsecond time scale. Existing experimental approaches, such as pulse radiolysis [44,45,46], are difficult to adapt to ultra-high dose rates, and commonly used methods, such as fluorescence and optical trailing techniques [47,48,49], typically focus on longer time scales, often in the millisecond range. These rely on detecting secondary reaction products like $H_{2}O_{2}$ formed after secondary reactions (e.g., using Amplex Red or coumarin acid), which are not suitable for the early stages of the chemical processes. In bridging simulations and experiments, we aim to expand TRAX-CHEM’s utility further by incorporating its analytical extension, TRAX-CHEMxt [40], to link dose and dose rate values to extended time-scale simulations. This will facilitate future comparisons with experimental data as measurement methodologies evolve to accommodate ultra-high dose rates. Preliminary estimates suggest that oxygen and scavengers exert a minor but notable influence under physioxic conditions, and these factors will be systematically integrated into future analyses. Despite the current lack of experimental data on sub-microsecond time scales, our findings demonstrate that the TRAX-CHEM framework provides reliable insights into ultra-high dose rate scenarios. By quantifying intertrack effects and their dependence on LET, we lay the groundwork for experimental validation once appropriate methodologies become available. In the interim, our results contribute to the broader understanding of track structure and chemical species evolution under ultra-high dose rate conditions.

## 4. Materials and Methods

### 4.1. Multiple Track Implementation

TRAX is a Monte Carlo track structure code [38] designed for precise modeling of low-energy electron and ion interactions on nanoscopic scales, with an energy range spanning from a few hundred MeV/u for ions and a few MeV for electrons down to 1–10 eV. Its single-interaction approach enables detailed simulations of ionization, excitation, and elastic scattering, supported by an adaptable database of target materials, including water, air, and plastics. These features make TRAX an ideal choice for simulating radiation effects in both dosimetric and biological systems, particularly where nanoscopic precision is critical. TRAXCHEM [39], an extension of TRAX that includes the chemical evolution, has been a pioneer in FLASH radiotherapy studies, being the first code to incorporate oxygenation effects [21] and extensions to homogeneous chemistry timescales [40].

In this study, we have further extended TRAX and TRAXCHEM to account for intertrack interactions, enabling the simulation of overlapping tracks up to the heterogeneous chemical stage (1 μs). The track evolution is modeled as a three-step process, according to the standard paradigm of radiation damage: physical, pre-chemical, and chemical stages. Each step is characterized by a distinct time scale. The basic version of the code (TRAX) addresses the physical stage and describes the ionization and excitation events, and the secondary electron production induced by radiation. This phase is supposed to last $10^{-15}$ s after the irradiation and provides the spatial distribution of the ionized and excited water molecules and sub-excitation electrons. Although, in principle, simultaneous multiple ionizations on the same water molecule could occur during this stage through inelastic interactions, the extremely short timeframe for physical collisions makes this scenario highly improbable in our context, besides at extremely high LET values. It is important to mention, moreover, that for all the cases studied, reflecting realistic, even if elevated, dose rates, the time interval between arrivals of different tracks exceeds femtoseconds making two tracks always practically independent for what concerns the physics phase, which conventionally ends after 1 fs from irradiation.

The following pre-chemical stage ($10^{-15}$–$10^{-12}$ s after irradiation), consists of the formation of the radiolytic chemical species originating from the dissociation processes of ionized and excited water molecules, and the thermalization and hydration of sub-excitation electrons. These events can be assumed to occur independently of those in the vicinity, allowing for the exclusion of possible intertrack effects at this stage [28].

During the final heterogeneous chemical stage, the generated species are tracked using a step-by-step algorithm, simulating Brownian diffusion and chemical reactions between themselves and the solvent for each time step until approximately $10^{-6}$ s after the irradiation when a more uniform condition is reached. Due to the wide time span covered by the simulation (6 orders of magnitude), a partial reduction in the computational time is achieved by gradually adjusting the time step during the simulation, increasing it from $10^{-13}$ s to $10^{-11}$ s. Indeed, while the time steps have to be selected small enough, to guarantee a precise sampling of the early stages of chemical evolution, characterized by rapid reaction kinetics, a larger time step can be employed for the latter part of track evolution when species have already diffused largely, and reaction rates are significantly lower. In this stage where the primary species are highly reactive and the chemical evolution is very dynamic, intertrack reactions are especially important and can affect the final, i.e., at 1 μs, radiolytic yields significantly.

From a computational standpoint, even when two tracks are simulated at a temporal separation of Δt, the physical and prechemical stages –where no intertrack interaction is assumed– are initiated simultaneously. The temporal delay of the second track is accounted for by activating the reactivity and diffusion of its chemical species when the time evolution of the first projectile reaches the time Δt.

At this point, the timestep is reset in order to sample the faster kinetics of the chemical reactions induced by the second particle, the species from both the first and second tracks are allowed to react with each other, and the simulation time is extended to 1 μs after the arrival time of the second track.

### 4.2. Calculation, Conditions, and Geometry

As mentioned in the introduction, we examined pairs of charged particle tracks separated by spatial and/or temporal intervals to evaluate the possible impact of intertrack reactions on the final radical and molecular yields. The projectiles used to investigate intertrack effects were: 20 MeV protons, 5 MeV/u helium ions, and 240 MeV/u carbon ions. Each of these corresponds to a distinct LET, determined in single-track simulations and tabulated in Figure 5. The choice of the carbon energy is related to the estimated energy in the target of the first in-vivo experimental evidence of a FLASH effect at high LET (carbon ion in entrance channel [50], representing an extreme case of verification. The analysis was carried out on a water volume of dimensions 5×5×Depth
μm^3^, where “Depth” denotes the length of the volume along the trajectory of the primaries. This quantity varies according to the primary LET to ensure the fulfillment of the track segment condition hypothesis, i.e., that the LET of the primary radiation remains almost constant within the simulation volume. As shown in Figure 5, the simulation target comprises three distinct water volumes: a main one where the evolution of the track over time is monitored, and two scatter volumes designed to guarantee the secondary particle equilibrium. These account for the build-up and the back-scattering of secondary particles, respectively. Their dimensions are sufficient to elicit the desired effects while maintaining the track segment condition.

We investigate various degrees of intertrack reactions for pairs of primaries, characterized by different combinations of spatial distances Δx = 0, 1, 10, $10^{2}$, and $10^{3}$ nm, along with temporal separations Δt = 1, 10, $10^{2}$ ps, 10, $10^{2}$, $3\cdot 10^{2}$, $7\cdot 10^{2}$, $10^{3}$ ns. In the simulations, ions were initiated with identical energy, and their trajectories were directed parallel to the positive z-axis. Specifically, the initial position of the first ion was, in (x,y,z) coordinates, $\left(\frac{\Delta x}{2},0,0\right)$, while the second ion was initiated at $\left(-\frac{\Delta x}{2},0,0\right)$ with a time delay of Δt (Figure 5).

For each simulation setup, a series of independent parallel calculations was carried out to enhance statistical significance. The number of simulations was chosen to be large enough to minimize substantial statistical fluctuations, effectively corresponding to a total energy deposition (number of independent simulation × LET × Depth) of 2 MeV in the main water box (see Appendix A).

The yield of different chemical products is described in terms of G-values, defined as the total number N(t) of species at a given time *t* produced or consumed per 100 eV of total deposited energy E(t) in the main volume(6)$$\mathrm{G-value}\left(t\right)=\frac{N(t)}{E(t)[100\;\mathrm{eV}]}.$$ While the energy is deposited in the physical stage only, the dependency of *E* on time arises due to the arrival of the second track, thus becoming a step function. Therefore, until the second projectile arrives, the G-value must be calculated using only the energy released by the first track.

## 5. Conclusions

This research revealed notable differences in the intertrack recombination probabilities among various particle types and LET values. Although overall intertrack effects were often negligible, significant deviations occurred in regions of close spatiotemporal proximity. While these effects diminished with increasing distances (both Δx and Δt), they persisted for some species, likely due to broader track structures. High LET radiation (helium and carbon) exhibited substantial enhancement of chemical reactions, where denser tracks influenced species production and depletion. Non-monotonic effects were observed, such as the behavior of $H_{2}O_{2}$, which decreases at higher Δt for carbon and helium, a trend previously seen only in experiments [47,49].

Our analysis revealed that intertrack effects involve more than just geometric considerations but complex spatiotemporal features. Future studies will account for oxygen and scavengers to model these effects with larger significance for biological systems, mimicking the environment, an essential aspect for accounting for intertrack effects in FLASH radiotherapy.

## Figures and Tables

**Figure 1 ijms-26-00571-f001:**
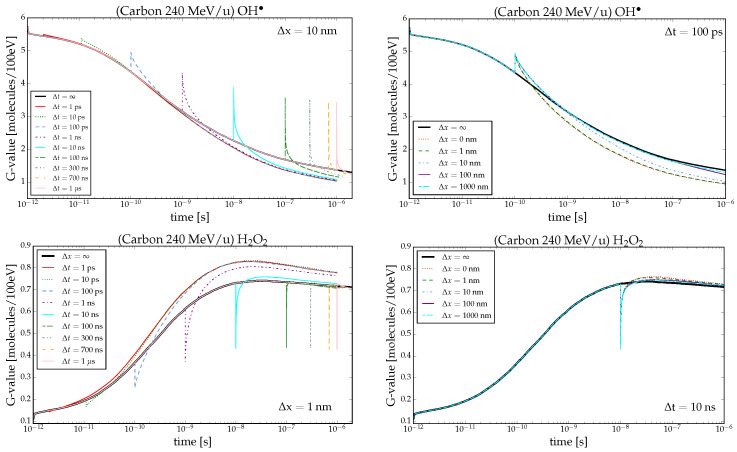
Time-dependent G-values in pure water calculated for 240 MeV/u carbon ions. In the left column, the G-values for $OH^{\cdot }$ and $H_{2}O_{2}$ are illustrated at fixed spatial track separation of Δx =10 nm and Δx =1 nm, respectively, and various Δt. In the right column, the G-values for the same chemical species are depicted at a fixed temporal distance of Δt =100 ps and Δt =10 ns, respectively, and different Δx. In all plots, Δx =∞ and Δt =∞ are represented by the black curve, indicating the NI scenario.

**Figure 2 ijms-26-00571-f002:**
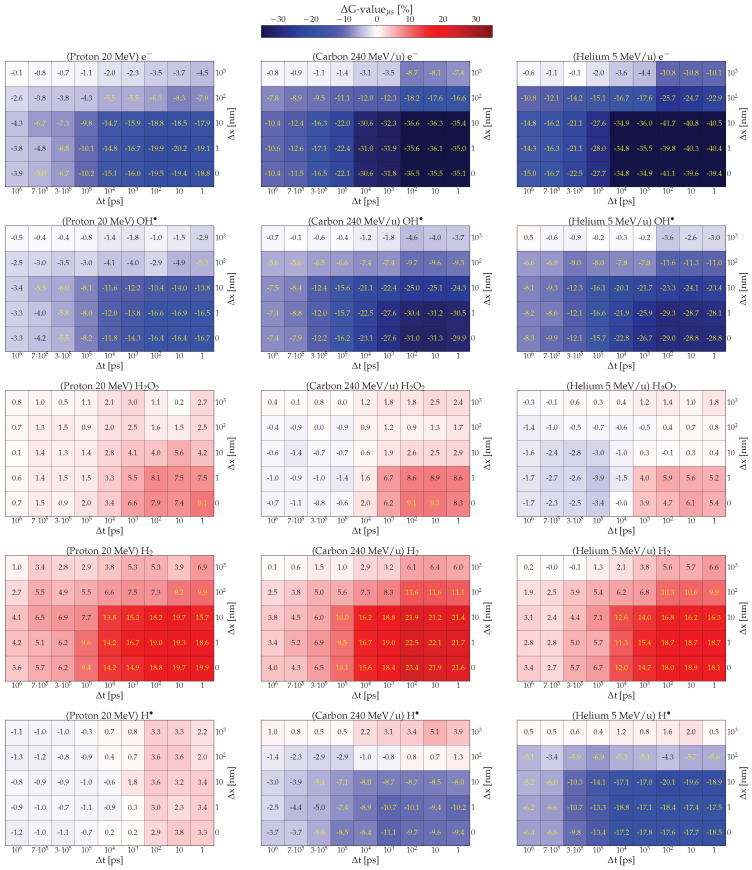
The $\Delta {\mathrm{G-value}}_{\mu s}$ (cf. Equation (Equation 2)) for five chemical species—$e^{-}$, $OH^{\cdot }$, $H_{2}O_{2}$, $H_{2}$, and $H^{\cdot }$—resulting from the interaction of two 20 MeV protons (first column), 240 MeV/u carbon ions (second column), and 5 MeV/u helium ions (third column) in pure water, is shown as a function of the spatial separation Δx and temporal separation Δt between tracks. Each bin represents the relative difference between the intertrack effect and the NI scenario.

**Figure 3 ijms-26-00571-f003:**
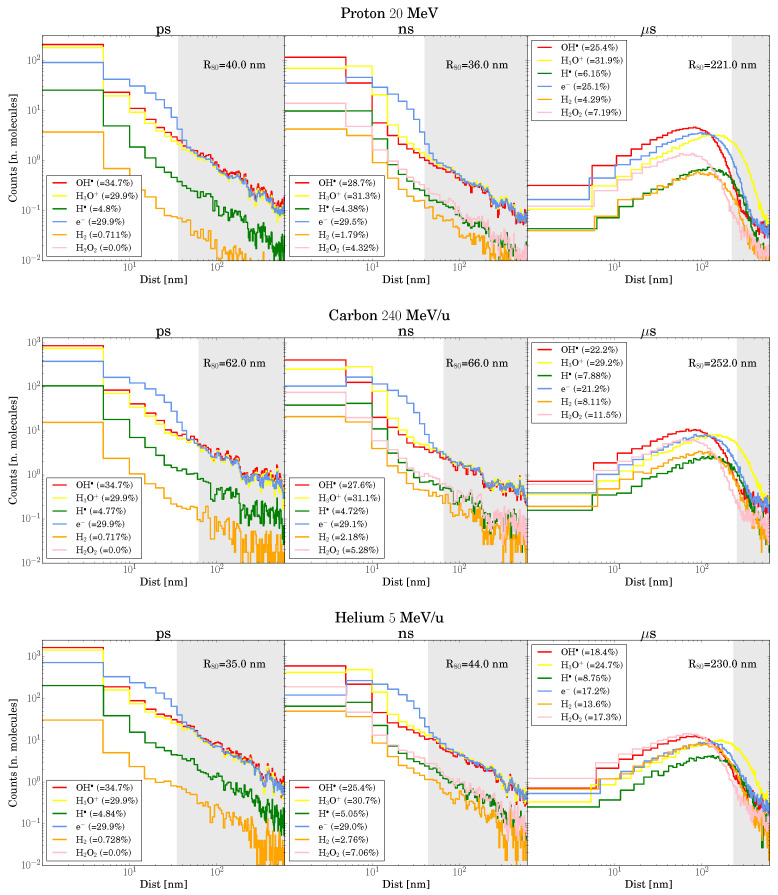
Each plot corresponds to a different primary particle and includes three panels showing the distribution of chemical species at various radial distances from the track center over different times. The legend on each plot indicates the percentage of each chemical species. The gray-shaded region in each subplot marks the area where the distance exceeds the corresponding $R_{80}$ (the radius containing 80% of chemical species) at a given time.

**Figure 4 ijms-26-00571-f004:**
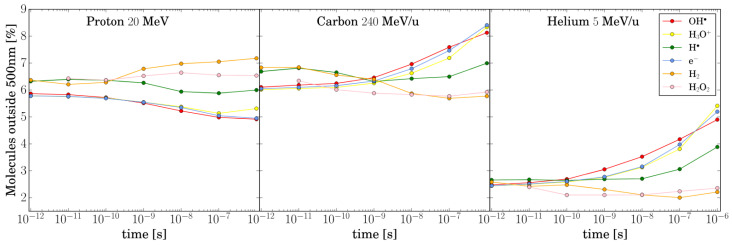
The percentage of molecules located beyond the 500 nm radius from the track center is calculated for protons, carbon ions, and helium ions. This percentage is determined relative to the total number of molecules of each type.

**Figure 5 ijms-26-00571-f005:**
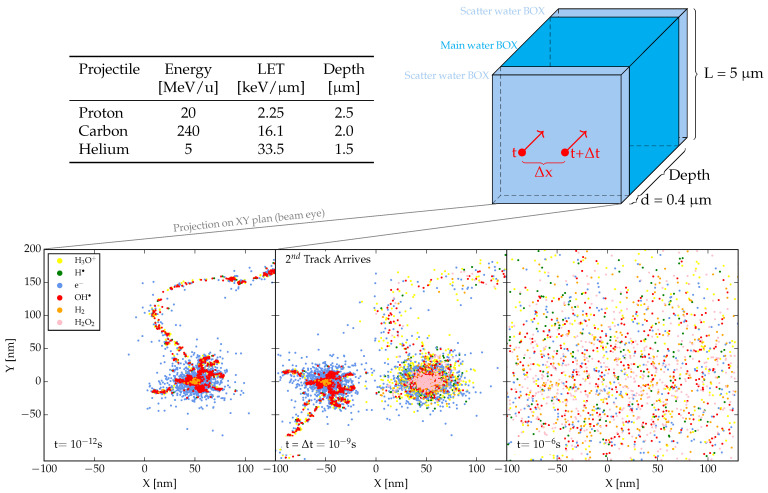
Schematic illustration of the target traversed by the primary particle. The volume (upper right) consists of three components, all made of water. The central parallelepiped represents the scoring volume, where the evolution of tracks over time is examined and monitored. The scatter volumes serve to describe the effect of secondary particles resulting from both back and forward scattering events. Detailed information on the primary particles simulated is provided in the table (upper left). At the bottom, a beam eye projection displays the primaries and their interactions within the scoring volume, where tracks arrive at varying times and positions (specified by Δx and Δt), and their chemical evolution is analyzed. The image specifically depicts the arrival and development of tracks generated by two helium ions at Δt =1 ns and Δx =100 nm.

## Data Availability

The data that support the findings of this study are available from the corresponding author upon reasonable request.

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
