# Peer review of "Probing Spatiotemporal Effects of Intertrack Recombination with a New Implementation of Simultaneous Multiple Tracks in TRAX-CHEM"

_ijms, 2025, doi:10.3390/ijms26020571_

Round 1
Reviewer 1 Report
Comments and Suggestions for Authors
Reference Report
Title: Probing spatiotemporal effects of intertrack recombination with a new implementation of simultaneous multiple tracks in TRAX-CHEM
Manuscript Number: IJMS-3349417
Submitted to: International Journal of Molecular Sciences
Authors: Castelli et al.
Evaluation Summary:
This manuscript employs Monte Carlo simulations to investigate the generation of reactive oxygen species (ROS) using a multiple-track model implemented in TRAX-CHEM. While the study presents an interesting approach, it has several critical issues that require clarification and improvement before publication.
Major Concerns
- Title and Relevance to FLASH Radiotherapy
- The manuscript focuses on the FLASH effect in radiotherapy, but the term “FLASH” is conspicuously absent from the title. Including "FLASH" in the title is essential to properly reflect the focus and significance of the study.
- Introduction
- The introduction lacks a concise overview of FLASH radiotherapy. A brief explanation of the FLASH effect, including its mechanisms and importance in radiotherapy, should be included. Relevant references, such as Cancers (2023, 15, 3883) and Encyclopedia (2023, 3, 808-823), can provide valuable context.
- The role and significance of ROS in FLASH radiotherapy, particularly its impact on DNA damage and cancer treatment, are inadequately addressed. The authors should discuss these aspects with supporting references such as Cells (2024, 13(10), 835).
- Rationale for Using TRAX-CHEM
- The authors should provide a detailed justification for selecting the TRAX-CHEM code, including a review of its suitability for simulating radiation tracks, linear energy transfer (LET), and cross-sections. Comparative discussions on its advantages and limitations would enhance the manuscript.
- Comparative Simulation Results
- To strengthen the credibility of their findings, the authors should include results obtained using other widely used simulation codes, such as GEANT4-DNA. This will allow readers to evaluate the reliability and robustness of the presented methodology.
- Simulation Parameters and Discussion
- The use of megavoltage particles in the simulations is questionable, as their attenuation over nanometer-scale distances is minimal. This limitation needs to be discussed thoroughly. Additionally, the paths or branches of ROS tracks should be elaborated upon in the context of the findings.
- Experimental Validation
- A major limitation of this work is the lack of experimental validation. Without experimental verification, the reliability and applicability of the simulation results remain unproven. The authors should either propose an experimental framework or discuss the challenges of conducting experimental validation in this context.
No comment.
Reviewer 2 Report
Comments and Suggestions for Authors
I am grateful for the opportunity to review manuscript # ijms-334917, entitled “Probing spatiotemporal effects of intertrack recombination with a new implementation of simultaneous multiple tracks in TRAX-CHEM.”
The authors have submitted a fascinating account of their analysis of the spatiotemporal effects of intertrack recombination for protons, helium-, and carbon-ions using TRAX-CHEM. Details of the modeled physical and chemical processes by TRAX and TRAX-CHEM, respectively, have been published previously.
This work is a step towards elucidating the mechanisms underlying the FLASH effect by modeling both ultra-high dose rates and various LETs, both of which impact the spatiotemporal proximity of multiple tracks. This study showed enhanced deviations from nominal chemical effects as a function of closer spatiotemporal proximity and provides the foundation for further investigation.
The paper is well written, and the data are clearly presented. This is good work, and I have no particular comments other than to point out minor typographical errors exist throughout that will need correcting during copyediting.
One such error in particular:
· Section 3.1 – second paragraph. Description of Fig. 2 has the assignment of fixed time delay and fixed spatial distance the wrong way round.
Author Response
[Comments] The paper is well-written, and the data are clearly presented. This is good work, and I have no particular comments other than to point out minor typographical errors exist throughout that will need correcting during copyediting.
One such error in particular:
- Section 3.1 – second paragraph. Description of Fig. 2 has the assignment of fixed time delay and fixed spatial distance the wrong way round.
[Reply] We are thankful to the reviewer for investing their time in assessing our paper, for the comments of appreciation of our research, and for the precious insights for corrections. We inserted the indicated correction and performed a careful spell and grammar check across the entire manuscript The modified parts in the text are marked in blue.
Round 2
Reviewer 1 Report
Comments and Suggestions for Authors
The authors have addressed my concerns.